# Identification and Expression Profiling of the 5-HT Receptor Gene in *Harmonia axyridis*

**DOI:** 10.3390/insects14060508

**Published:** 2023-05-31

**Authors:** Qiqi Zhang, Yifang Chang, Changying Zheng, Lijuan Sun

**Affiliations:** Research Center for Environment-Friendly Agricultural Pest Management, College of Plant Medicine, Qingdao Agricultural University, Qingdao 266109, China; qqzhang77@sina.com (Q.Z.); changyifang0927@163.com (Y.C.); zhengcy67@qau.edu.cn (C.Z.)

**Keywords:** *Harmonia axyridis* (Pallas), 5-HT receptor, gene cloning, expression profile

## Abstract

**Simple Summary:**

*Harmonia axyridis* (Pallas) (Coleoptera: Coccinellidae) is a natural predatory enemy insect widely distributed in north China and has been widely used in agricultural practice. Owing to a low daily average temperature from December to February of the following year in greenhouses and the facultative diapause of adult *H. axyridis*, the application of this ladybird in greenhouses in north China is limited. It is of great importance to regulate its predatory ability to improve its biological control efficiency in greenhouses. The 5-hydroxytryptamine (5-HT) was found to play a regulatory role in the predation of *H. axyridis* in our previous study, but the 5-HT receptor of *H. axyridis* has not been clearly identified so far, and its expression characteristics are poorly understood. In this study, the 5-HT receptor gene of *H. axyridis* was cloned and identified by phylogenetic tree construction and multiple sequence alignments. The expression patterns of each receptor in different developmental stages and tissues were analyzed by quantitative real-time PCR (qRT-PCR). The results showed that *H. axyridis* expressed four 5-HT receptor subtypes, named *5-HT_1A_Har*, *5-HT_1B_Har*, *5-HT_2_Har,* and *5-HT_7_Har*. All 4 receptors were expressed at high levels in the adult stage, especially in 2-day-old adults, with expression levels of 18.72-fold (male) and 14.21-fold (female) of that in eggs for *5-HT_1A_Har*, 32.27-fold (male) and 83.58-fold (female) of that in eggs for *5-HT_1B_Har*, 36.82-fold (male) and 119.35-fold (female) of that in eggs for *5-HT_2_Har*, and 165.47-fold (male) and 115.59-fold (female) of that in eggs for *5-HT_7_Har*. The level of expression for each receptor gene decreased with the advance of day-age in adults. The levels of expression of *5-HT_1B_Har*, *5-HT_2_Har,* and *5-HT_7_Har* were low at the egg, larval, and pupal stages, and *5-HT_1A_Har* was not expressed in the larval stage. The four receptors were expressed in the nervous system, digestive tract, pectoral muscles, and male and female gonads of adults. The *5-HT_1A_Har* was expressed at a high level in the pectoral muscle (6.75-fold of that in the nervous system), *5-HT_1B_Har* in male gonads (1.02-fold of that in the nervous system) and the nervous system, *5-HT_2_Har* in male gonads (5.74-fold of that in the nervous system), and *5-HT_7_Har* in the digestive tract (1.81-fold of that in the nervous system). The results of this study will lay a foundation for research on the function of the 5-HT receptor by RNA interference in the regulation of predation by *H. axyridis*.

**Abstract:**

It has been found that 5-hydroxytryptamine (5-HT) modulates the feeding of some insects, and this phenomenon was found in *Harmonia axyridis* (Pallas) by our previous study. An understanding of the 5-HT system in this beetle is helpful for utilizing 5-HT to modulate its predation to improve biological control efficiency, especially in greenhouses in winter in north China. This is because 5-HT influences diapause in insects by modulating the synthesis and release of prothoracic hormone (PTTH) and, therefore, influences feeding. To elucidate the molecular basis of the *H. axyridis* 5-HT system, reverse-transcription polymerase chain reaction (RT-PCR), multiple sequence alignment, and phylogenetic tree construction were used to identify the 5-HT receptor in *H. axyridis*, and quantitative real-time PCR (qRT-PCR) was used to analyze the expression pattern of these receptor genes in different developmental stages and in the nervous system (brain + ventral nerve cord), digestive tract, pectoral muscles, and gonads of the adult ladybird. The results showed that four 5-HT receptors were identified in *H. axyridis*, named *5-HT_1A_Har*, *5-HT_1B_Har*, *5-HT_2_Har,* and *5-HT_7_Har*. The four receptors were expressed at high levels in the adult stage, especially in 2-day-old adults, with expression levels of 18.72-fold (male) and 14.21-fold (female) of that in eggs for 5-HT_1A_, 32.27-fold (male) and 83.58-fold (female) of that in eggs for 5-HT_1B_, 36.82-fold (male) and 119.35-fold (female) of that in eggs for 5-HT_2_, and 165.47-fold (male) and 115.59-fold (female) of that in eggs for 5-HT_7_. The level of expression decreased with the advance of day-age in adults. The levels of expression of *5-HT_1B_Har*, *5-HT_2_Har,* and *5-HT_7_Har* were low at the egg, larval, and pupal stages, and *5-HT_1A_Har* was not expressed in the larval stage. The four receptors were expressed in the nervous system, digestive tract, pectoral muscles, and male and female gonads. The *5-HT_1A_Har* was expressed at a high level in the pectoral muscle (6.75-fold of that in the nervous system), *5-HT_1B_Har* in male gonads (1.02-fold of that in the nervous system) and the nervous system, *5-HT_2_Har* in male gonads (5.74-fold of that in the nervous system), and *5-HT_7_Har* in the digestive tract (1.81-fold of that in the nervous system). The results of this study will lay a foundation for research on the function of the 5-HT receptor by RNA interference in the regulation of predation by *H. axyridis*.

## 1. Introduction

The 5-hydroxytryptamine (5-HT), also known as serotonin, is an ancient and conserved small biological molecule widely distributed in nature. As a neurotransmitter, which is a chemical that transmits information between neurons or between neurons and effector cells such as muscle cells, gland cells, etc., 5-HT controls and regulates various important physiological activities of organisms including humans, nematodes, and insects [1]. The 5-HT in insects can regulate growth and development, reproduction [2,3,4,5], diapause [6], feeding [7,8], learning and memory [9], behavior [10,11,12], olfaction [13], circadian rhythm [14,15,16], immune response [17], movement [18], cardiac rhythm regulation [19], and gregarization [20], etc. Studying the 5-HT of insects is necessary to reveal the behavior and physiological mechanisms of insects, which are commonly the basis of beneficial insect utilization and pest control.

The 5-HT works primarily by acting on specific receptors. The 5-HT receptor is a phylogenetically ancient receptor that evolved 750 million years ago and is present in invertebrates and higher mammals [21]. At present, 5-HT receptors have been cloned in more than 10 species of insects; most of these receptors belong to the G-protein-coupled receptor superfamily [22]. G-protein-coupled receptors (GPCRs) are a receptor protein superfamily that contains seven α-helical transmembrane (TM) structures. GPCRs activate the second messenger pathway to transfer ligand signals into cells to regulate specific physiological processes via conjugated heterotrimer G proteins (α, β, γ subunit) [23,24]. GPCRs are involved in chemo- and photoreception, hormonal physiology, synaptic function, and a variety of other processes [25]. Three types of 5-HT receptor have been cloned in insects, namely 5-HT_1_, 5-HT_2_, and 5-HT_7_ [22]. In most insects, there are two subtypes of 5-HT_1_, namely 5-HT_1A_ and 5-HT_1B_; there are also two subtypes of 5-HT_2_, namely 5-HT_2A_ and 5-HT_2B_ [22,26,27]. Qi et al. [28] identified a novel 5-HT_8_ receptor without homology with vertebrate 5-HTs in *Pieris rapae.* Li et al. [29] found the 5-HT_1D_ subtype of 5-HT_1_ and the 5-HT_7A_ and 5-HT_7B_ subtypes of 5HT_7_ in *Aedes aegypti.*

*Harmonia axyridis* (Pallas) is a highly adaptable, natural predatory enemy insect widely distributed in north China. It has a strong ability to prey on aphids, *Tetranychus* mites, scale insects, and the eggs of some Lepidopteran pests. As an important natural enemy for biological control, it has been widely used in agricultural production in China [30]. Owing to a low daily average temperature from December to February of the following year in greenhouses and the facultative diapause of adult *H. axyridis*, the application of this ladybird in greenhouses in north China is limited, for the food intake by ladybirds in diapause is low. It is of great importance to regulate its predatory ability to improve its biological control efficiency in greenhouses. Wang et al. [6] found that 5HTRB locked the gate of PTTH release and synthesis in the Chinese silk moth, *Antheraea Pernyi,* and led to a delay of emergence of adults. We suspect that the 5-HT pathway may be involved in control of diapause in *H. axyridis*, and, therefore, influences its predation. A previous injection experiment showed that 5-HT plays an obvious regulatory role in the predation of *H. axyridis* (unpublished data), so it is important to study the 5-HT receptors of *H. axyridis*; but the 5-HT receptor of *H. axyridis* has not been clearly identified so far, and its expression characteristics are poorly understood. In this study, the 5-HT receptor gene of *H. axyridis* was cloned and identified by phylogenetic tree construction and multiple sequence alignments. The expression patterns of each receptor in different developmental stages and tissues were analyzed by quantitative real-time PCR (qRT-PCR). The purpose of this study was to lay a foundation for the functional study of the 5-HT receptor in *H. axyridis*.

## 2. Materials and Methods

### 2.1. Insects

The *H. axyridis* eggs were collected from peach trees on the campus at Qingdao Agricultural University in April 2019, and fed on *Myzus persicae* after hatching to establish a laboratory colony. The laboratory colony was established at 20 ± 1 °C and 65 ± 10% relative humidity under a photoperiod of 12 h light:12 h dark in a manual climatic box (RGN-300, Ningbo Southeast Instrument Co., Ltd., Ningbo City, China). *M. persicae* were cultivated on Chinese cabbage seedlings under the same conditions of temperature, light, and humidity in another manual climatic box. The *H. axyridis* used for experiments were collected from the laboratory colony.

Virgin *H. axyridis* adults at 14 days old were collected for 5-HT receptor cloning. Twenty eggs at one-day-old; fifteen first instar larvae at two-day-old; ten second instar larvae at two-day-old; five third instar larvae at two-day-old; two fourth instar larvae at two-day-old; two pupae at two-day-old; two virgin females at two-day-old, seven-day-old and fourteen-day-old; and two virgin males at two-day-old, seven-day-old, and fourteen-day-old were collected for analysis of the expression pattern of 5-HT receptor genes in different developmental stages. Five to ten virgin adults at sixteen-day-old were collected and anesthetized on ice to isolate the nervous system (brain + ventral nerve cord), digestive tract, pectoral muscles, and gonads for analysis of the expression pattern of the 5-HT receptor gene in different tissues; each tissue was removed to an Rnase-free Eppendorf centrifuge tube charged with ice-cold 0.9% normal saline. All the samples were frozen in liquid nitrogen and quickly transferred into a freezer at −80 °C for later use.

### 2.2. The mRNA Isolation and cDNA Synthesis

The samples were fully ground in liquid nitrogen, and total RNA was isolated with TRIzol Reagent (Invitrogen, Carlsbad, CA, USA). The RNA isolated from each sample was dissolved in DEPC-treated water (Sangon Biotech, Shanghai, China), and the quality of the RNA was assessed on a 1% agarose gel, and quantities were determined on a spectrophotometer (Nano Photometer TM-Class, Implen GmbH, Munich, Germany). The isolated RNA with A260/A280 values ranging from 1.8 to 2.0 was used for cDNA synthesis. Single-strand cDNA was synthesized from 3μL RNA of 200 μg/μL using a ReverTra Ace-a-kit (TaKaRa, Dalian, China).

### 2.3. Cloning of the Serotonin Receptor Gene from H. axyridis

According to the predicted 5-HT receptor gene open reading frame (ORF) of *H. axyridis* in the InsectBase (www.insect-genome.com accessed on 9 November 2021), primers were designed using Primer 5.0 (the base sequences of the primers are listed in Table 1). PCR reactions were carried out with a 2× Spark FastHiFi PCR Master Mix kit (Qingdao Haosail Science Co., Ltd., Qingdao, China) according to the manufacturer’s instructions and consisted of one cycle at 94 °C for 3 min, 40 cycles at 94 °C for 30 s, 50 °C (for *Haxy004670.1*) or 52 °C (for *Haxy019702.1* and *Haxy005697.1*) or 55 °C (for *Haxy019754*) for 30 s, and 72 °C for 30 s, with an out-of-cycle extension step of 72 °C for 10 min. The annealing temperatures for each gene are listed in Table 1. The PCR product was separated by electrophoresis on a 1.0% agarose gel, and the purified product was sent to Sangon Biotech Co., LTD. (Shanghai, China) for sequencing using a 3730 XL DNA analyzer (Applied Biosystems, Carlsbad, CA, USA).

### 2.4. Multiple Sequence Alignment and Phylogenetic Analysis

The derived amino acid sequences were used for phylogenetic analysis. The phylogenetic tree and molecular evolutionary analyses were performed using MEGA 7.0 software with the neighbor-joining method [31]. The accession numbers of the 5-HT receptor genes of other insect species are listed in Table 2. Multiple sequence alignments were identified by BLAST programs from the NCBI (http://blast.ncbi.nlm.nih.gov/Blast.cgi accessed on 11 October 2022). Multiple sequence alignments of the complete amino acid sequences were perform ed with ClustalX (http://www.genome.jp/tools-bin/clustalw accessed on 11 October 2022). The ENDscript/ESPript website (http://espript.ibcp.fr/ESPript/cgi-bin/ESPript.cgi accessed on 12 October 2022) was used to map the sequence alignment results. The transmembrane segments were predicted with Deep TMHMM 2.0 (https://dtu.biolib.com/DeepTMHMM accessed on 7 March 2023).

### 2.5. Quantitative Real-Time PCR (qRT-PCR)

According to the target genes sequence obtained in Section 2.3, primers for qRT-PCR were designed using Primer 5.0, and the 18sRNA (F: ACGGACTTCGGTAGGACG; R: CGCAGACAATCCCGAAA) gene was used as the reference gene for qRT-PCR. The base sequences of the primers are listed in Table 3. The qRT-PCR reactions were carried out with a Spark Taq PCR Master Mix kit (Qingdao Haosail Science Co., Ltd., Qingdao, China) according to the manufacturer’s instructions, and consisted of 1 cycle at 94 °C for 180 s, 40 cycles of 94 °C for 15 s, 55 °C (for *Haxy019702.1*, *Haxy004670.1* and *Haxy005697.1*) or 60 °C (for *Haxy019754*) for 15 s, and 72 °C for 20 s. The annealing temperatures for each gene are listed in Table 3. There were three biological replicates for each treatment and three technical replicates for each biological replicate.

### 2.6. Statistical Analysis

Relative quantitative data were calculated according to the 2^−ΔΔCt^ method [32]. The expression level of the 5-HT receptor genes at the egg stage was set to be 1, and those in different developmental stages were compared with those in the egg stage. The expression level of 5-HT receptor genes in the nervous system was set to be 1, and those in different tissues were compared with those in the nervous system. Data were analyzed by one-way ANOVA followed by Tukey’s multiple comparison test. Statistical significance was associated with values of *p* < 0.05.

## 3. Results

### 3.1. Cloning of the 5-HT Receptor Gene

The electrophoretogram of cloned products was shown in Figure 1. A *Haxy004670.1* fragment of 772 bp, *Haxy005697.1* fragment of 834 bp, *Haxy019702.1* fragment of 1061 bp, and *Haxy019754* fragment of 1053 bp were cloned, respectively.

### 3.2. Multiple Sequence Analysis of 5-HT Receptor Genes

The amino acid sequences deduced from the *Harmonia* 5-HT receptor genes were compared with those of other insects, and the result indicated that the *Harmonia* 5-HT receptor was structurally similar to the 5-HT receptors of other known insects, all with similarity exceeding 50%. Haxy004670.1 was 68.06% identical to the 5-HT_1A_ receptors of *Tribolium castaneum*. Haxy005697.1 was 78.06% and 66.00% identical to the 5-HT_1B_ receptors of *T. castaneum* and *A. pernyi*, respectively. Haxy019702.1 was 65.48% and 59.57% identical to the 5-HT_2A_ receptors of *Drosophila melanogaster* and *Apis mellifera*, respectively. Haxy019754 was 78.70% and 55.70% identical to the 5-HT_7_ receptors of *Aedes aegypti* and *A. mellifera*, respectively.

The cDNA fragment from Haxy004670.1 obtained through gene cloning encoded three transmembrane (TM) segments corresponding to the fifth to seventh TM segments of GPCRs (TM5, TM6, and TM7 in Figure 2A). The molecular phylogenetics of 5-HT_1A_ receptors were analyzed using the deduced amino acid sequence of the partial cDNA fragment of Haxy004670.1 and those of the corresponding part of various 5-HT receptors in other species; the results indicated that Haxy004670.1 was closely related to the insect 5-HT_1A_ proteins (Figure 3A).

The cDNA fragment from Haxy019754 obtained through gene cloning encoded three transmembrane (TM) segments, corresponding to the fourth to sixth TM segments of GPCRs (TM4, TM5, and TM6 in Figure 2B). The molecular phylogenetics of 5-HT_1B_ receptors were analyzed using the deduced amino acid sequence of the partial cDNA fragment of Haxy005697.1 and those of the corresponding parts of various 5-HT receptors in other species; the results indicated that Haxy005697.1 was closely related to the insect 5-HT_1B_ proteins (Figure 3A).

The cDNA fragment of Haxy019702.1 obtained through gene cloning encoded two transmembrane (TM) segments corresponding to the fourth to sixth TM segments of GPCRs (TM6 and TM7 in Figure 2C). The molecular phylogenetics of 5-HT_2_ receptors were analyzed using the deduced amino acid sequence of the partial cDNA fragment of Haxy019702.1 and those of the corresponding parts of various 5-HT receptors in other species; the results indicated that Haxy019702.1 was closely related to the insect 5-HT_2_ proteins (Figure 3B).

The cDNA fragment of Haxy019754 obtained through gene cloning encoded two transmembrane (TM) segments corresponding to the fifth to seventh TM segments of GPCRs (TM5, TM6, and TM7 in Figure 2D). The molecular phylogenetics of 5-HT_7_ receptors were analyzed using the deduced amino acid sequence of the partial cDNA fragment of Haxy019754 and those of the corresponding parts of various 5-HT receptors in other species; the results indicated that Haxy019754 was closely related to the insect 5-HT_7_ proteins (Figure 3C).

By multiple sequence alignment and construction of a phylogenetic tree, *Haxy004670.1* was identified as *5-HT_1A_*, *Haxy005697.1* was identified as *5-HT_1B_*, *Haxy019702.1* was identified as *5-HT_2_*, and *Haxy019754* was identified as *5-HT_7_*. According to the sequence similarity and the current nomenclature rules of 5-HT receptors, each newly identified gene was named after the receptor type plus genus name [33], so they were named *5-HT_1A_Har* (GenBank accession number: OQ724534), *5-HT_1B_Har* (GenBank accession number: OQ724535), *5-HT_2_Har* (GenBank accession number: OQ724536), and *5-HT_7_Har* (GenBank accession number: OQ724536).

### 3.3. Expression Pattern of 5-HT Receptor Genes in Different Developmental Stages in H. axyridis

The expression patterns of the 5-HT receptor genes in different developmental stages were investigated using RT-qPCR analysis. The following twelve developmental stages, egg, first instar larva, second instar larva, third instar larva, fourth instar larva, pupa, two-day-old adult, seven-day-old adult, and fourteen-day-old adult were subjected to RT-qPCR analysis. As shown in Figure 4, the expression profiles of 5-HT receptor genes in different developmental stages of *H. axyridis* are generally consistent with only slight differences. All receptor genes were expressed at high levels in the adult stage; the levels of expression in the 2-day-old and 7-day-old adults were significantly higher than that in the preadult stage. In the adult stage, the level of expression decreased with age, and a sharp drop was found in the 14-day-old female adults. The *5-HT_1A_Har* was not expressed in the larval stage and had its highest level of expression in 2-day-old adult males (18.72-fold of that in eggs), differing from that in 2-day-old females (14.21-fold of that in eggs). The highest levels of expression of *5-HT_1B_Har* and *5-HT_2_Har* were both recorded in 2-day-old female adults (83.58-fold of that in eggs for *5-HT_1B_Har* and 119.35-fold of that in eggs for *5-HT_2_Har*), and both differed from their expression in 2-day-old male adults (32.27-fold of that in eggs for *5-HT_1B_Har* and 36.82-fold of that in eggs for *5-HT_2_Har*). The highest expression level of *5-HT_7_Har* was recorded in 2-day-old male adults (165.47-fold of that in eggs), which differed from that in female adults (115.59-fold of that in eggs). The level of expression of *5-HT_7_Har* in the larval stage increased with the increase in instar, and the level of expression in the fourth instar larvae was highest (24.70-fold of that in eggs), differing from that in other instars (2.86~10.14-fold of that in eggs). The levels of expression of all the 5-HT receptors genes were low in the pupal stage.

### 3.4. The Expression Pattern of 5-HT Receptor Genes in Different Tissues in H. axyridis

As shown in Section 3.3, the level of expression of 5-HT receptor genes in *H. axyridis* is highest at the adult stage. Therefore, adult *H. axyridis* were collected for analysis of the expression pattern of 5-HT receptor genes in different tissues. Five tissues of the ladybird including the nervous system, digestive tract, pectoral muscles, female gonads and male gonads were subjected to RT-qPCR analysis. As shown in Figure 5, the highest level of expression of *5-HT_1A_Har* was recorded in the pectoral muscles (6.75-fold of that in the nervous system), followed by the male gonads (2.66-fold of that in the nervous system), nervous system, digestive tract (0.60-fold of that in the nervous system) and female gonads (0.04-fold of that in the nervous system), and the level of expression in different tissues differed. The highest level of expression of *5-HT_1B_Har* was recorded in the male gonads (1.02-fold of that in the nervous system) and the nervous system (the levels of expression did not differ from each other), followed by the digestive tract (0.08-fold of that in the nervous system), female gonads (0.12-fold of that in the nervous system), and pectoral muscles (0.16-fold of that in the nervous system), and the level of expression in the digestive tract did not differ from that in the female gonads. The highest level of expression of *5-HT_2_Har* was recorded in the male gonads (5.74-fold of that in the nervous system), followed by the female gonads (1.71-fold of that in the nervous system), nervous system, pectoral muscles (0.27-fold of that in the nervous system), and digestive tract (0.23-fold of that in the nervous system), and the levels of expression in the last two tissues did not differ from each other. The highest level of expression of *5-HT_7_Har* was recorded in the digestive tract (1.81-fold of that in the nervous system), followed by the male gonads (1.06-fold of that in the nervous system) and the nervous system, the pectoral muscle (0.63-fold of that in the nervous system) and the female gonads (0.05-fold of that in the nervous system), and the level of expression in the nervous system did not differ from that in the male gonads.

## 4. Discussion

In this study, four 5-HT receptor genes, *5-HT_1A_Har*, *5-HT_1B_Har*, *5-HT_2_Har,* and *5-HT_7_Har*, were identified in *H. axyridis*. Sequence alignment revealed that the amino acid residues of the 5-HT receptor were conserved among different insect species. Phylogenetic analysis showed that the same 5-HT receptor family subtypes in different organisms clustered preferentially in one clade, followed by 5-TH receptors in different species clustered in different clades based on genetic distance.. The same family of receptors first clustered into one clade, suggesting that 5-HT_1_, 5-HT_2_, and 5-HT_7_ receptors were formed before the divergence of vertebrates from invertebrates. Each receptor type was differentiated into different subtypes; for example, insect 5-HT_1_ diverged into 5-HT_1A_ and 5-HT_1B_, and 5-HT_2_ diverged into 5-HT_2A_ and 5-HT_2B_. These isoforms of the 5-HT receptor each clustered into a small clade that may have developed independently in vertebrates and invertebrates.

It was shown that 5-HT receptor genes were expressed differently in different developmental stages of *H. axyridis*. Furthermore, *5-HT_1B_Har*, *5-HT_2_Har,* and *5-HT_7_Har* were expressed in all the developmental stages of *H. axyridis*, while *5-HT_7_Har* was expressed at all stages except larva. The levels of expression of the four receptors in the adult stage were significantly higher than those in the preadult stage; this discovery coincides with that of Wang et al. [33] in their study of expression patterns of four 5-HT receptors (5-HT_1A_, 5-HT_1B_, 5-HT_2A_, and 5-HT_2B_) in different developmental stages in *Helicoverpa armigera* (Hübner). Four 5-HT receptor genes in *H. axyridis* were expressed at high levels in the adult stage of *H. axyridis*, indicating that these receptors may be involved in physiological functions in adults. The *5-HT_1A_Har* was not expressed in the larval stage but only in the adult stage and the developmental stage adjacent to the adult, indicating that this gene may be related to physiological phenomena specific to the adult. The expression of *5-HT_7_Har* in the larval stage of *H. axyridis* showed an upward trend with the increase in instar and reached the highest level at the fourth instar. The studies of Zhao [34] and Yu [35] all showed that daily predation by *H. axyridis* larvae was greatest at the fourth instar, so we suspected that the expression of *5-HT_7_Har* was related to predation. The 5-HT_7_ receptor affects feeding by mediating gastrointestinal smooth muscle relaxation in mammals [36], and this phenomenon was also found in the honeybee, *Apis mellifera mellifera* [8]. However, the relationship between the expression of *5-HT_7_Har* and predation by *H. axyridis* needs further study. Moreover, 5-HT innervation and 5-HT receptors were found in the alimentary tract of *A. mellifera mellifera* [8], *Leucophaea maderae* [37], *Camponotus mus* (Roger) [38], and *Locusta migratoria* (Fabricius) [39]. A study of the distribution of 5-HT innervation and 5-HT receptors may be helpful for understanding the mechanism by which 5-HT regulates predation by *H. axyridis*. It was also found in our study that the expression of four 5-HT receptor genes in *H. axyridis* decreased with the advance of adult day-age, indicating that these 5-HT receptors were related to some age-dependent physiological functions of adults. The levels of expression of *5-HT_1A_Har* and *5-HT_7_Har* were significantly higher in males than in females, while the levels of expression of *5-HT_1B_Har* and *5-HT_2_Har* were higher in females; these phenomena suggested that the expression of these genes may be associated with sex-related physiological functions.

The expression patterns of 5-HT receptors in different tissues can help generate hypotheses related to their biological significance in insects. Tissue localization of 5-HT receptors has been reported in a variety of insects and arthropods. *H. armigera* (Hübner) *5-HT_1_* (including 5-HT_1A_ and 5-HT_1B_) [33], *Gryllus bimaculatus 5-HT_1B_* [40], and *Bombyx mori 5-HT_1_* [41] were all reported to be expressed at high levels in the nervous system but expressed at low levels or not expressed in the intestinal tract of *H. armigera*. In this study, we found that *5-HT_1A_ Har* was expressed at a high level in the pectoral muscle and *5-HT_1B_ Har* in the nervous system, but both showed low expression in the digestive tract; this result was generally consistent with that found in the above three insects. In addition, 5-HT_2_ receptors have been found to be expressed at high levels in the salivary glands of *Calliphora vicina* and *G. bimaculatus*, and the level of expression in the brain of *C. vicina* is relatively high [26,40]. In this study, the level of expression of the *5-HT_2_ Har* receptor in the nervous system was lower than that in the gonads, which was inconsistent with the expression pattern in *G. bimaculatus* and *C. vicina*. Furthermore, *5-HT_7_ Har* is reported to be expressed at a high level in the midgut of *G. bimaculatus* [41] and *Eriocheir sinensis* [42], which is consistent with our findings in *H. axyridis*. In this study, the *Harmonia* 5-HT receptor was found to be expressed in various tissues at differential expression levels, suggesting that the 5-HT system is widely distributed in this ladybird.

The results for the expression profile of the *H. axyridis* 5-HT receptors in this article will provide a basis for the functional study of them, and further studies on expression sites, timing of expression, pharmacological reaction and demographical effect, etc., are needed to elucidate the exact function of these receptors.

## Figures and Tables

**Figure 1 insects-14-00508-f001:**
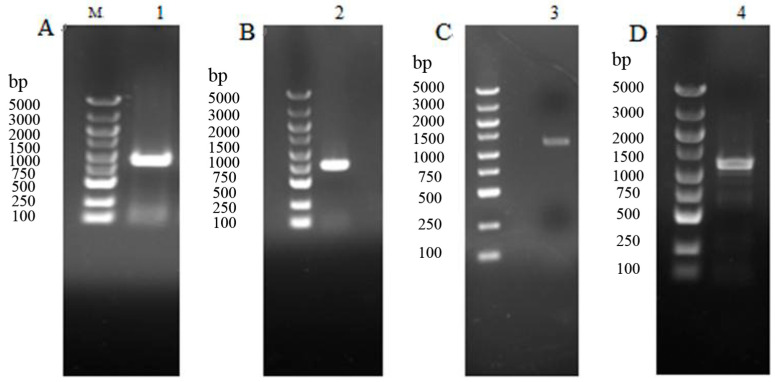
Electrophoretogram of PCR products of 5-HT receptor genes in *Harmonia axyridis*. M: BM 5000+ marker, lanes 1–4: PCR amplification products of *Haxy004670.1*, *Haxy005697.1*, *Haxy019702.1*, and *Haxy019754*. (**A**) PCR products of *Haxy004670.1*. (**B**) PCR products of *Haxy005697.1*. (**C**) PCR products of *Haxy019702.1.* (**D**) PCR products of *Haxy019754*.

**Figure 2 insects-14-00508-f002:**
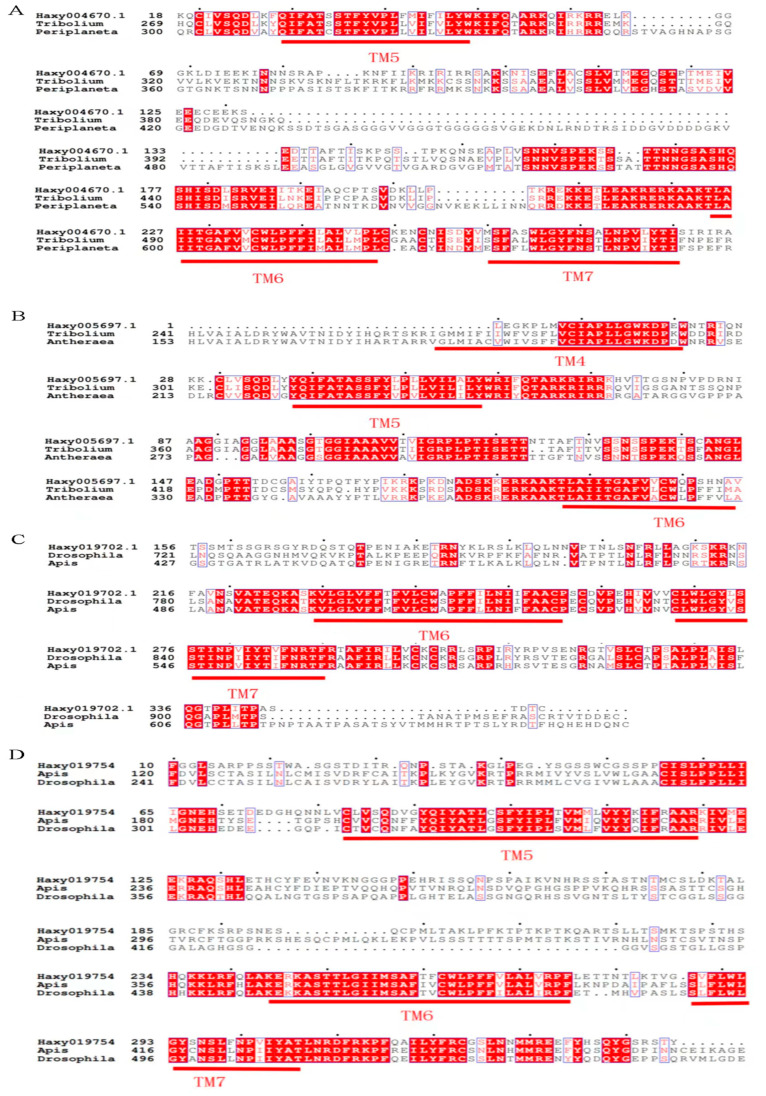
Amino acid sequence alignment of 5-HT receptors from *Harmonia axyridis* and other insect species. (**A**) Alignment against sequences of Haxy004670.1 with 5-HT_1A_ receptors from *T. Castaneum* and *P. americana*. (**B**) Alignment against sequences of Haxy005697.1 with 5-HT_1B_ receptors from *T. Castaneum* and *A. pernyi*. (**C**) Alignment against sequences of Haxy019702.1 with 5-HT_2A_ receptors from *D. melanogaster* and *A. mellifera.* (**D**) Alignment against sequences of Haxy019754 with 5-HT_7_ receptors from *D. melanogaster* and *A. mellifera.* The alignment was generated using ClustalX alignment software. Putative transmembrane domains are indicated by red bars (TM1-7). Identical residues between the receptors are shown as white characters against a red background. Conservatively substituted residues are framed in blue. The accession numbers of 5-HT in relevant species in GenBank are listed in Table 2.

**Figure 3 insects-14-00508-f003:**
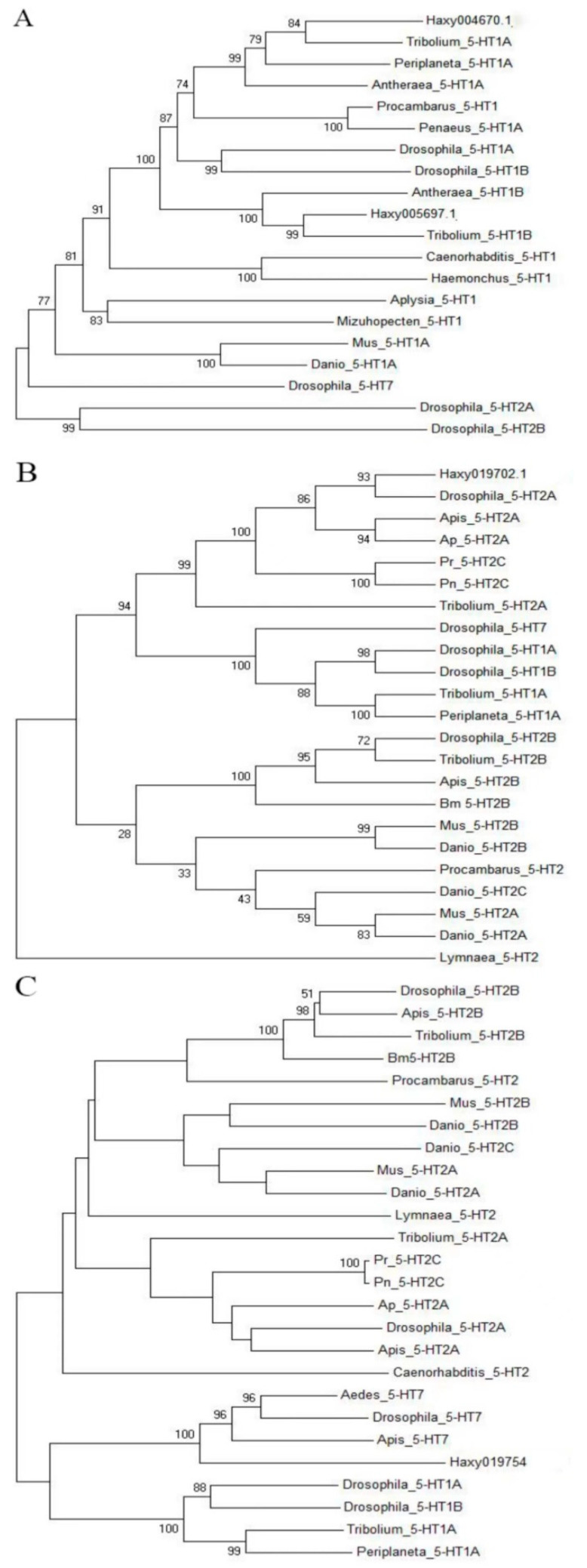
Phylogenetic tree based on amino acid sequences of 5-HT receptors of *Harmonia axyridis* and other species (1000 replicates). The GenBank accession numbers of the proteins used for phylogenetic analysis are listed in Table 2. (**A**) Phylogenetic tree for *Haxy004670.1*. and *Haxy005697.1*. (**B**) Phylogenetic tree for *Haxy019702.1.* (**C**) Phylogenetic tree for *Haxy019754*.

**Figure 4 insects-14-00508-f004:**
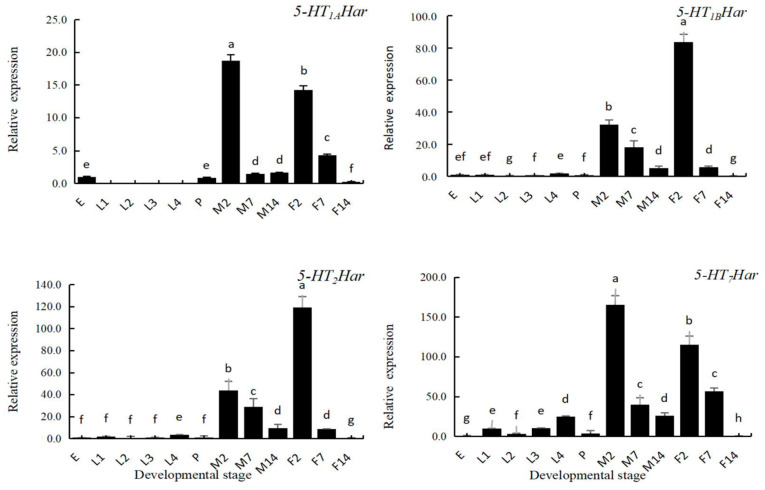
The expression pattern of four 5-HT receptor genes in different developmental stages in *Harmonia axyridis* by qPCR. Abbreviations: egg (E), first instar larvae (L1), second instar larvae (L2), third instar larvae (L3), fourth instar larvae (L4), pupa (P), 2-day-old-male (M2), 7-day-old male (M7), 14-day-old male (M14), 2-day-old female (F2), 7-day-old female (F7), 14-day-old female (F14). The expression level of 5-HT receptor genes at egg stage was set to be 1, and those in different developmental stages were compared with that in egg stage. Different lowercase letters on the columns indicate significant differences in gene expression between different stages. (*p* < 0.05, Tukey’s test).

**Figure 5 insects-14-00508-f005:**
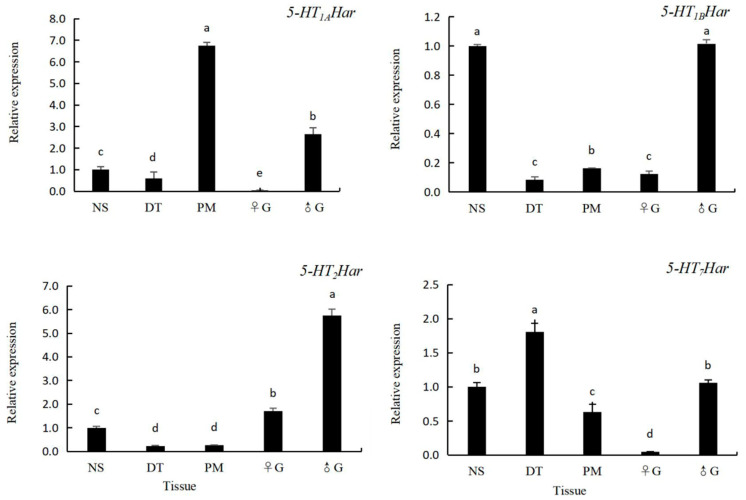
The expression pattern of four 5-HT receptor genes in different tissues in 14-day-old adult *Harmonia axyridis* by qPCR. Abbreviations: nervous system (NS), digestive tract (DT), pectoral muscles (PM), female gonads (♀G), and male gonad (♂G). The expression level of 5-HT receptor genes in the nervous system was set to be 1, and those in different tissues were compared with that in the nervous system. Different lowercases letters on the columns indicate significant differences between expression levels in different tissues. (*p* < 0.05, Tukey’s test).

**Table 1 insects-14-00508-t001:** Primers used for gene cloning.

Gene	Primer Name	Primer Sequence (5′-3′)	Annealing Temperature/°C
*Haxy019702.1*	02F	AGGTACATTGTTGAGTTGCAGC	52
	02R	TAAATGGAGGAGGCAGTAGAGA	
*Haxy019754*	54F	CACGAAAAAGGGTAGCCAGCA	55
	54R	CGCACTCCCACCAGAAGAAGC	
*Haxy004670.1*	70F	AAAGGCAACCAACAAACTACAAA	50
	70R	CCAAGAGAGAGAAGAAAGAGACG	
*Haxy005697.1*	97F	GAGGGCTGCCAACAGACCACGAA	52
	97R	CGCTGACTCAAAGAAGGAACGGA	

**Table 2 insects-14-00508-t002:** Accession numbers of 5-HT receptors in different insect species.

Receptor	Name	Species	Accession Number
5-HT_1_	Drosophila 5-HT_1A_	*Drosophila melanogaster*	NP_476802
	Antheraea 5-HT_1A_	*Antheraea pernyi*	ABY85410
	Tribolium 5-HT_1A_	*Tribolium castaneum*	XP_967449
	Periplaneta 5-HT_1A_	*Periplaneta americana*	CAX65666
	Drosophila 5-HT_1B_	*Drosophila melanogaster*	NP_523789
	Antheraea 5-HT_1B_	*Antheraea pernyi*	ABY85411
	Tribolium 5-HT_1B_	*Tribolium castaneum*	XP_972856
	Procambarus 5-HT_1_	*Procambarus clarkii*	ABX10973
	Penaeus 5-HT_1_	*Penaeus monodon*	AAV48573
	Caenorhabditis Ser-4	*Caenorhabditis elegans*	NP_497452
	Haemonchus 5-HT_1_	*Haemonchus contortus*	AAO45883
	Aplysia 5-HT_ap1_	*Aplysia californica*	AAC28786
	Mizuhopecten 5-HT_1_	*Mizuhopecten yessoensis*	BAE72141
	Mus 5-HT_1A_	*Mus musculus*	NP_032334
	Danio 5-HT_1A_	*Danio rerio*	NP_001116793
5-HT_2_	Drosophila 5-HT_2A_	*Drosophila melanogaster*	NP_730859
	Apis 5-HT_2A_	*Apis mellifera*	NP_001189389
	Tribolium 5-HT_2A_	*Tribolium castaneum*	XP_972327
	Drosophila 5-HT_2B_	*Drosophila melanogaster*	NP_649806
	Apis 5-HT_2B_	*Apis mellifera*	NP_001191178
	Tribolium 5-HT_2B_	*Tribolium castaneum*	EFA04642
	Procambarus 5-HT_2_	*Procambarus clarkii*	ABX10972
	Caenorhabditis 5-HT_2_	*Caenorhabditis elegans*	NP_001024728
	Mus 5-HT_2A_	*Mus musculus*	NP_766400
	Mus 5-HT_2B_	*Mus musculus*	NP_032337
	Danio 5-HT_2A_	*Danio rerio*	CAQ15355
	Danio 5-HT_2B_	*Danio rerio*	NP_001038208
	Pr 5-HT_2C_	*Pieris rapae*	XP_022122944.2
	Pn 5-HT_2C_	*Pieris napi*	XP_047521401.1
	Bm5-HT_2B_	*Bombyx mori*	XP_021208983.1
	Danio 5-HT_2C_	*Danio rerio*	NP_001123365
	Ap 5-HT_2A_	*Agrilus planipennis*	XP_018318891.1
	Lymnaea 5-HT_2_	*Lymnaea stagnalis*	AAC16969
5-HT_7_	Drosophila 5-HT_7_	*Drosophila melanogaster*	NP_524599
	Aedes 5-HT_7_	*Aedes aegypti*	AF296125
	Apis 5-HT_7_	*Apis mellifera*	NP_001071289
	Caenorhabditis (Ser-7)	*Caenorhabditis elegans*	NP_741730

**Table 3 insects-14-00508-t003:** Primers for fluorescent quantitative PCR reaction.

Gene	Primer Name	Primer Sequence (5′-3′)	Annealing Temperature/°C
*Haxy019702.1*	02F	AGGTACATTGTTGAGTTGCAGC	55
	02R	TAAATGGAGGAGGCAGTAGAGA	
*Haxy019754*	54F	CACGAAAAAGGGTAGCCAGCA	60
	54R	CGCACTCCCACCAGAAGAAGC	
*Haxy004670.1*	70F	AAAGGCAACCAACAAACTACAAA	55
	70R	CCAAGAGAGAGAAGAAAGAGACG	
*Haxy005697.1*	97F	GAGGGCTGCCAACAGACCACGAA	55
	97R	CGCTGACTCAAAGAAGGAACGGA	

## Data Availability

The data presented in this study are available upon request from the corresponding author.

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
