# Peer review of "Identification and Expression Profiling of the 5-HT Receptor Gene in Harmonia axyridis"

_insects, 2023, doi:10.3390/insects14060508_

Round 1

Reviewer 1 Report

Neurotransmision in insects is the key to implement an effective
chemical control and to understand the physiological mechanisms, but
this research targets 5HT receptors without elucidating the cellular
structure using 5HT neural network, serotonin and 5HTR antagonists/
agonists effects. We must study these preliminary aspects before asking
for receptor structures and functions. A plenty of literature is
available for 5HT functions such as Wang et al., 2013, PLoSONE. After
that, RNAi should be tried to examine if the inference to a particular function. Rather, the current study should focus on the
structural aspects such as seven membrane pass domains, ion-channel
and phosphorylation sites etc., I would suggest.

I can understand the text but the formality is not cared much.

Author Response

Dear reviewer:

        We appreciate the time and effort that you have dedicated to providing your valuable feedback on my manuscript. Our response to your comment was as follow.

Point 1: Neurotransmision in insects is the key to implement an effective chemical control and to understand the physiological mechanisms, but this research targets 5HT receptors without elucidating the cellular structure using 5HT neural network, serotonin and 5HTR antagonists/agonists effects. We must study these preliminary aspects before asking for receptor structures and functions. A plenty of literature is available for 5HT functions such as Wang et al., 2013, PLoSONE. After that, RNAi should be tried to examine if the inference to a particular function. Rather, the current study should focus on the structural aspects such as seven membrane pass domains, ion-channel and phosphorylation sites etc., I would suggest.

Response 1: Thank you for your comments and good suggestions. This study aimed to elucidate the molecular basis of the H. axyridis 5-HT system and provide basis for further study on function of 5HTR. We have studied the effect of 5-HT on predation of H. axyridis (the data is unpublished) and this work was the basis and reason we concern 5HTR of H. axyridis. We are sorry not explaining it clearly in the introduction. We have revised the introduction according to your suggestions. We are sorry for neglecting an important literature, it was retrieved and read intensively, and quoted in the manuscript. Thank you for your good suggestions, to study function of 5HTR in H. axyridis by RNAi and structure of 5HTR in H. axyridis is our next task. Your suggestion is helpful for us, thank you again!

Point 2: L7: Family, order name!

Response 2: Thank you for your comments and suggestion. We have supplemented the family and order name.

Point 3: L12: present perfect cannot use previously that indicates past

Response 3: Thank you for your comments. We are sorry for our mistakes and have corrected it.

Point 4: L16: Scientific results must be explained on the basis of experimental design, rationale, and the results. Need a brief description of the data.

Response 4: Thank you for your comments and good suggestion. We gave a brief description of the data and explained the scientific result in the section “Simple Summary”. Is it OK? If there is any question, please tell us, thank you!

Point 5: L19: why?

Response 5: Thank you for your comments. We have revised the background in the abstract to explain why it is important to study the 5-HT receptor gene of H. axyridis.

Point 6: L68: Study on insect 5HTRs has not been extensively reviewed. In relationship to diapause regulation, one critical case is missing, that is the Antheraea pernyi case.

The reason why you suspected that 5HTR regulation is critical in diapause regulation has not been given. Introduction is very weak.

Response 6: Thank you for your comments and direction. We retrieved this literature about Antheraea pernyi and made an intensive reading. We are sorry for missing it. We have quoted it in the section “introduction” to explain the importance of study on 5HTR of H. axyridis. Thank you again.

Point 7: L71: The laboratory colony of. H. axyridis was established from the field sample collected where at what stage and when? Geographic origin and host plant must be described.

Response 7: Thank you for your comments and good suggestion. We have supplemented the information about collection time, site, and stage of H. axyridis, as well as the method for laboratory colony establishment. Thank you!

Point 8: L74: Myzus percicae is not something called foodsfuff. Prey is better.

Response 8: Thank you for your comments and good suggestion. We avoided the improper word “foodsfuff” in the revised version. Thank you!

Point 9: L77: adults

Response 9: Thank you for your comments, we have corrected it.

Point 10: L227: in which tissue?

Response 10: Thank you for your comments. We don’t know whether our presentation made a confusion, if it is, we are sorry. The result presented here was a result for spatiotemporal expression of 5-HT receptor genes in H. axyridis, aimed to explain why we used adults as material to study the tissue expression of 5-HT receptor genes. We revised the sentence, if there is any question, please tell us, thank you!

Point 11: L230-232 spaciotemporal patterns not shown. We cannot attribute the change to particular functions!

Response 11: Thank you for your comments. Here, we still think previous presentation “As shown in Figure 4, the level of expression of 5-HT receptor genes in H. axyridis is highest at the adult stage.” at the beginning of this paragraph made a confusion, please let us say sorry again. This paragraph is for tissue expression of 5-HT receptor genes, the insect used for RNA extraction was 14 days old adults, it is difficult to give the spaciotemporal patterns. Furthermore, we are sorry for not understanding your comments about “We cannot attribute the change to particular functions!” It may refer to the improper expression in “Discussion”? We have revised “Discussion”, thank you!

Point 12: L243: which tissue?

Response 12: Thank you for your comments. We are sorry because we think the word used in the little title was not consistent with that in the caption confused you. Figure 4 is for spatiotemporal expression of 5-HT receptor genes in H. axyridis, the whole body of this beetle was used for RNA isolation, so it is difficult for us to tell which tissue it is.

Point 13: L249: at what stage? which cell tipes?

Response 13: Thank you for your comments. We are sorry not presenting the stage of adult in the captain of figure 5, we have supplemented it. We are sorry for not understanding your comment about “which cell tipes?” Would you please interpret it? Thank you!

Point 14: L267: This is a sloppy inference to physiological function.

Response 14: Thank you for your comments. We are sorry for making a less rigorous inference to physiological function. We have deleted this sentence.

Point 15: L281: I cannot be not convinced.

Response 15: Thank you for your comments. We are sorry for mistyped the word “suspect” to be “suggest”. The function of 5-HT7Har can only be predicted according the result of our current work, RNAi should be tried to examine the exact functions.

Point 16: L283: Has someone demonstrated the presence of 5HT innervation in the gastroenteric muscles of insects?

Response 16: Thank you for your comments and question. 5HT innervation was found in alimentary tract of honeybee, Leucophaea maderae, Camponotus mus (Roger), Locusta migratoria (Fabricius), but no evidence indicates its existing in gastroenteric muscles of insects. We revised section “Discussion” and supplemented sone discussion relavent to it.

Reviewer 2 Report

The authors conducted effective experiments to identify the 5-hydroxytryptamine genes of Harmonia axyridis. These results provide a valuable foundation for future investigations into the physiological functions and behavioral activities of H. axyridis. However, there are some points that could be improved, which are outlined below.

L19. The abstract could benefit from a brief introduction that explains why understanding the 5-HT system in H. axyridis is important. This will help readers understand the significance of the study and why it was conducted.

L35. The introduction could benefit from a sentence or two that explains why it is important to study 5-HT in insects. For instance, what insights can be gained from understanding its functions in this group of organisms?

L37. It could be helpful to define “neurotransmitter” for readers who may not be familiar with the term.

6. The sentence "5-HT in insects can regulate a variety of physiological functions and behavioral activities" could be rephrased to make it clear that this is not limited to insects only, as the previous sentence mentioned the importance of 5-HT in organisms in general.

L47. Provide a brief explanation of what G-protein-coupled receptors are and how they function, as this could be helpful for readers who are not familiar with this type of receptor.

L59. Suggestion for improvement is to provide a brief explanation that how diapause affects the behavior of H. axyridis.

L87. Authors should provide more information about the methodology for mRNA isolation and cDNA synthesis, such as the amount of material used for RNA extraction, the concentration and quality of the resulting cDNA. Additionally, it is advisable to mention the annealing temperature used during PCR reactions to provide more detail on the experimental conditions.

L125. The annealing temperature range for different genes (55°C-60°C) is quite broad. It would be useful to specify the exact annealing temperature used for each gene.

L139. Authors should provide the information that which software was used to conduct the statistical analysis.

L169. The legend of fig.2 should be together in the same page with chart.  

Author Response

Dear reviewer:

      We appreciate the time and effort that you have dedicated to providing your valuable feedback on my manuscript. Our responses to your comment are as follow.

Point 1: L19. The abstract could benefit from a brief introduction that explains why understanding the 5-HT system in H. axyridis is important. This will help readers understand the significance of the study and why it was conducted.

Response 1: Thank you for your comments and good suggestion. We explained why understanding the 5-HT system in H. axyridis is important in the section “abstract.

Point 2: L35. The introduction could benefit from a sentence or two that explains why it is important to study 5-HT in insects. For instance, what insights can be gained from understanding its functions in this group of organisms?

Response 2: Thank you for your comments and good suggestion. We have explained why it is important to study 5-HT in insects in section “Introduction”, thank you!

Point 3: L37. It could be helpful to define “neurotransmitter” for readers who may not be familiar with the term.

Response 3: Thank you for your comments and good suggestion. We have supplemented the concept of “neurotransmitter” to provide better understanding to readers who may not be familiar with the term.

Point 4: 6. The sentence "5-HT in insects can regulate a variety of physiological functions and behavioral activities" could be rephrased to make it clear that this is not limited to insects only, as the previous sentence mentioned the importance of 5-HT in organisms in general.

Response 4: Thank you for your comments and good suggestion. The sentence "5-HT in insects can regulate a variety of physiological functions and behavioral activities" have been rephrased to be “5-HT controls and regulates various important physiological activities of organisms including humans, nematodes and insects.”

Point 5: L47. Provide a brief explanation of what G-protein-coupled receptors are and how they function, as this could be helpful for readers who are not familiar with this type of receptor.

Response 5: Thank you for your comments and good suggestion. We revised manuscript and provided a brief explanation of what G-protein-coupled receptors are and how they function according your suggestion.

Point 6: L59. Suggestion for improvement is to provide a brief explanation that how diapause affects the behavior of H. axyridis.

Response 6: Thank you for your comments and good suggestion. A brief explanation that how diapause affects the behavior of H. axyridis was provided.

Point 7: L87. Authors should provide more information about the methodology for mRNA isolation and cDNA synthesis, such as the amount of material used for RNA extraction, the concentration and quality of the resulting cDNA. Additionally, it is advisable to mention the annealing temperature used during PCR reactions to provide more detail on the experimental conditions.

Response 7: Thank you for your comments and good suggestion, we have provided the information you suggested in the section “Materials and method”. And we specified the exact annealing temperature used for each gene in the text in detail.

Point 8: L125. The annealing temperature range for different genes (55°C-60°C) is quite broad. It would be useful to specify the exact annealing temperature used for each gene.

Response 8: Thank you for your comment. The annealing temperatures for different genes are listed in Table 3. We are sorry for not specifying the exact annealing temperature used for each gene in the text in detail, we have supplemented them in the text.

Point 9: L169. The legend of fig.2 should be together in the same page with chart. 

Response 9: Thank you for your comment. We have revised the size of the figure and made the legend of fig. 2 be together in the same page with chart.

Round 2

Reviewer 1 Report

I have read the two abstracts but still found problems as enclosed. It is still incomplete for publication, though it is much better than before. Please ask the authors one mote time to revise it with care. English needs professional help by native speaker.

Please ask the authors one mote time to revise it with care. English needs professional help by native speaker.

Author Response

Dear reviewer,

    We appreciate the time and effort that you have dedicated to reviewing our manuscript and your valuable feedback, and our response is as follow.

Point 1: L17. temporal, yes but where is spacial?

Response 1: Thank you for your comments. The spatiotemporal expression pattern of one gene in insect was commonly composed by expression pattern in different stages and tissue, and the spacial expression pattern commonly refers to the tissue-specific expression pattern, so we are sorry for the improper expression. We have revised all the related expression in the full text, thank you

Point 2: L17-L18. where do you show tissue-specific expression pattern?

Response 2: Thank you for your comments. We studied the expression pattern of 4 receptor genes in different tissue, we are sorry for the improper expression here. We have revised all the related expression in the full text, thank you

Point 3: L23-L24. data not shown.

Response 3: Thank you for your comments and direction. We have supplemented the main data.

Point 4: L24-L26. data not shown.

Response 4: Thank you for your comments and direction. We have supplemented the main data.

Point 5: L26-L27. what functions are under serotonin control?

Response 5: Thank you for your comments and direction. The predation of Harmonia axyridis are under serotonin control, we revised the sentence to give the information in detail.

Point 6: L28. It has been found that 5HT modulates

Response 6: Thank you for your comments and help. We have revised the sentence according to your suggestion, thank you!

Point 7: L30. Effective

Response 7: Thank you for your comments and help. We have revised the word “effectively” to be “effective”, thank you!

Point 8: L30-L32. feeding, diapause or PTTH release, after all?

Response 8: Thank you for your comments and direction. It is feeding. We have revised the sentence. Thank you!

Point 9: L36. where?

Response 9: Thank you for your comments and direction. We revised the sentence and provide the tissues name in detail, thank you!

Point 10: L41-L43.where is the data?

Response 10: Thank you for your comments. We supplemented the data.

Point 11: L44-L45.how?

Response 11: Thank you for your comments. RNA interference is preferred to study the function of 5-HT receptors in H. axyridis. We have supplied the method in the sentence.

Thank you and best regards.

Yours sincerely,

Lijuan Sun
